# HiFi-Foley: Multimodal Diffusion with Representation Alignment for High-Fidelity Foley Audio Generation

## Abstract

Recent advances in video generation produce visually realistic content, yet the absence of synchronized audio severely compromises immersion. To address key challenges in video-to-audio generation, including multimodal data scarcity, modal semantic response imbalance, and limited audio quality in existing methods, we propose HiFi-Foley, an end-to-end text-video-to-audio framework that synthesizes high-fidelity audio precisely aligned with visual dynamics and semantic context. Our approach incorporates three core innovations: (1) a novel multimodal diffusion transformer that addresses semantic response imbalance between video and text modalities through dual-stream audio-video fusion via joint attention and balanced textual semantic injection via cross-attention; (2) a representation alignment training strategy that employs self-supervised audio features to guide latent diffusion training, thereby improving audio quality and semantic consistency; (3) a scalable data pipeline leveraging open-source tools for cleaning raw data and constructing training datasets. Extensive evaluations demonstrate that HiFi-Foley achieves state-of-the-art performance across audio fidelity, visual-semantic alignment, temporal alignment, and distribution matching.

## 1 Introduction

Recent advances in video generation models (Polyak et al., 2025; Gao et al., 2025; Kong et al., 2025) have achieved notable success in synthesizing high-quality, photorealistic dynamic sequences. However, the absence of synchronized audio in these generated videos significantly undermines immersion. Traditional Foley art requires meticulous frame-by-frame creation by professionals, incurring substantial time and financial costs that render it incompatible with the efficiency of modern video generation systems. To address this limitation, research on automated Foley generation has gained momentum.

Text-to-audio (TTA) synthesis constitutes an early approach to Foley generation, producing high-quality audio conditioned exclusively on textual descriptions. The state-of-the-art (SOTA) TTA methods can produce high-fidelity audio well-aligned with semantic descriptions. Nevertheless, restricted to textual guidance only, TTA methods cannot inherently generate audio aligned with video content, which is a critical requirement for Foley generation.

Video-to-Audio (V2A) generation aims to produce high-quality audio precisely synchronized with video, both semantically and temporally. Recent V2A approaches (Cheng et al., 2025; Liu et al., 2025) based on the Multimodal Diffusion Transformer (MMDiT) framework have shown significant progress. These methods leverage dual-modal inputs (video and text), utilizing pre-trained encoders to extract video features and text embeddings to guide audio synthesis through diffusion or flow-matching processes. However, existing V2A methods suffer from several key limitations. (1) **Multimodal Data Scarcity**: Public datasets like VGGSound (Chen et al., 2020) offer only 556 hours of low-quality video-audio pairs, while high-quality TTA datasets (e.g., AudioCaps (Kim et al., 2019), WavCaps (Mei et al., 2024a)) lack video modality. The scarcity of multimodal data fundamentally limits the generalization capabilities of existing Text-Video-to-Audio (TV2A) models. (2) **Modality Imbalance**: Current methods exhibit over-reliance on text semantics, maintaining only coarse temporal alignment with video while inadequately responding to visual semantics. For

instance, when processing text "the sound of ocean waves" alongside video depicting a beach scene with people, seagulls and waves, the model exclusively generates the wave sounds while neglecting other audio elements (footstep sounds and seagull calls). This phenomenon demonstrates an imbalanced multimodal integration where textual cues dominate the audio generation at the expense of visual information. (3) **Audio Quality**: The fidelity of audio generated by existing methods fails to meet professional standards, exhibiting background noise and semantically inconsistent artifacts.

To overcome these challenges, we propose HiFi-Foley, an end-to-end multimodal TV2A generation model capable of synthesizing high-quality audio tightly aligned with both input video and text semantics. Our model adopts a multimodal flow-matching transformer paradigm trained on a large-scale text-video-audio multimodal dataset. First, to enable scalable multimodal dataset creation, we introduce a comprehensive data pipeline for automated labeling and filtering of collected data. This pipeline facilitated the construction of a 122k hours TV2A dataset. Second, to address modality imbalance, we propose a novel multimodal audio generation architecture comprising dual-stream MMDiT blocks and single-stream audio DiT blocks. The MMDiT incorporates joint self-attention with interleaved RoPE to strengthen temporal dependencies between video and audio, followed by the injection of textual information through the cross-attention mechanism. Third, we introduce a Representation Alignment (REPA) loss to enhance audio quality by aligning the hidden embeddings from the single-stream audio DiT block with the audio features extracted by a pre-trained self-supervised model (Li et al., 2023). Our key contributions are summarized as follows:

- We introduce HiFi-Foley, a novel TV2A framework that generates high-quality, semantically and temporally aligned audio from video and text inputs. Our approach mitigates modal semantic response imbalance, significantly enhancing visual-semantic alignment while sustaining text-semantic alignment, achieving SOTA performance.
- We introduce a REPA training strategy leveraging pre-trained audio features to provide semantic and acoustic guidance for the audio modeling process, effectively enhancing audio generation quality and semantic consistency.
- We propose an efficient TV2A data pipeline built on open-source tools for cleaning raw data and constructing large-scale, high-quality datasets.

## 2 RELATED WORK

**Text-to-Audio.** Early audio synthesis focuses on TTA generation, which aims to synthesize audio content based on textual descriptions. DiffSound (Yang et al., 2023) pioneers diffusion models for environmental sound synthesis. AudioGen (Kreuk et al., 2023) adopts auto-regressive transformer to predict discrete audio representation. Subsequent advances including AudioLDM (Liu et al., 2023), Make-An-Audio (Huang et al., 2023), and Stable Audio Open (Evans et al., 2024) utilize latent diffusion with text embeddings from pre-trained text encoders to enhance semantic alignment. Recently, TangoFlux (Hung et al., 2025) introduces a hybrid DiT architecture following Flux (Labs et al., 2025), with preference optimization, enabling high-fidelity TTA generation at reduced latency. However, TTA approaches are inherently limited to text-based generation and lack the capability to produce audio that is aligned with video content.

**Video-to-Audio.** Video-to-Audio synthesis aims to generate audio semantically and temporally consistent with video content. Existing V2A approaches can be broadly categorized into two paradigms, injecting visual features into pre-trained TTA models and training V2A models from scratch. In the first category, T2AV (Mo et al., 2024) introduces an Audio-Visual ControlNet to strengthen visual consistency in TTA models. FoleyCrafter (Zhang et al., 2024) utilizes semantic adapters and temporal controllers for alignment, injecting textual and visual embeddings into a UNet backbone through cross-attention to guide audio generation. VATT (Liu et al., 2024) uses text prompts derived from video content to steer TTA models for audio synthesis. For approaches trained from scratch, Diff-Foley (Luo et al., 2023) utilizes a contrastive audio-visual pre-training (CAVP) module to align features across modalities. FoleyGen (Mei et al., 2024b) employs autoregressive transformers to achieve visual feature-based audio generation. Recent works have demonstrated remarkable advances in both audio quality and multimodal alignment. Frieren (Wang et al., 2025) proposes an efficient V2A model based on rectified flow matching. MMAudio (Cheng et al., 2025) adopts a hybrid architecture combining MMDiT blocks with single-modality DiT blocks, incorporating synchronization features via Synchformer (Iashin et al., 2024), which is validated for temporal

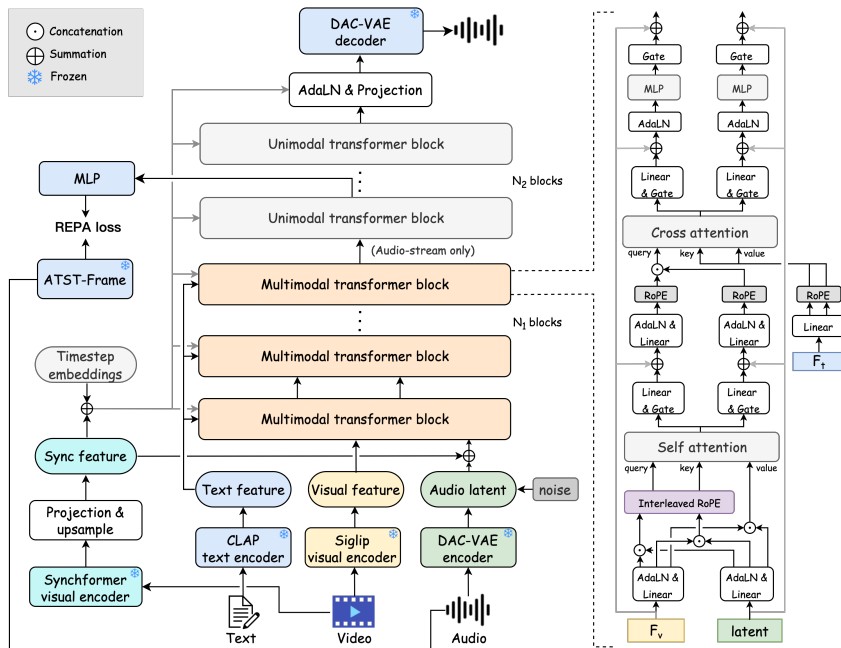

Figure 1: Overview of the HiFi-Foley model architecture. The proposed model integrates encoded text (CLAP), visual (SigLIP-2), and audio (DAC-VAE) inputs through a hybrid framework with $N_1$ multimodal transformer blocks followed by $N_2$ unimodal transformer blocks. The hybrid transformer blocks are modulated and gated with Synchformer synchronization features and timestep embeddings. A pre-trained ATST-Frame is used to compute REPA loss with mapped latent representations from a unimodal transformer block. The generated audio latent are decoded into audio waveforms by the DAC-VAE decoder.

alignment efficacy in V-AURA (Viertola et al., 2024). MMAudio achieves high-quality synthesis with enhanced alignment. Concurrent work ThinkSound (Liu et al., 2025) proposes a Chain-of-Thought (CoT) framework enabling step-by-step interactive audio generation and editing. While previous approaches have made significant progress in V2A synthesis, several critical challenges remain unresolved. These include suboptimal audio quality that falls short of professional standards, imprecise temporal alignment, and insufficient semantic correspondence with visual context. In contrast to existing methods, our approach employs distinct attention mechanisms to address the different alignment relationships between video-audio and text-audio modalities. This framework significantly enhances both video-semantic alignment and the quality of synthesized audio.

**Representation Alignment.** Representation Alignment (REPA), first introduced by (Yu et al., 2024), accelerates convergence and enhances semantic fidelity in large-scale generative models through aligning internal features with representations extracted from a pre-trained visual encoder. The REPA framework has since been widely adopted across generative modeling tasks. VA-VAE (Yao et al., 2025) integrates REPA into LightningDiT to improve variational autoencoder latent space learning. JanusFlow (Ma et al., 2025b) employs REPA for multimodal framework refinement; UniTok (Ma et al., 2025a) applies REPA to develop unified visual tokenizers; and MergeVQ (Li et al., 2025) utilizes REPA for vector quantization based model optimization. Building upon these successes, we apply REPA to TV2A synthesis, where we align intermediate representations of DiT blocks with frame-level audio features extracted from a pre-trained self-supervised model (Li et al., 2023) to enhance semantic and acoustic modeling. Our experimental results demonstrate marked improvements in both audio fidelity and semantic relevance.

## 3 HiFi-Foley

To achieve modality balance and high-quality TV2A generation, we introduce the HiFi-Foley framework. As illustrated in Figure 1, HiFi-Foley employs a hybrid architecture with $N_1$ multimodal

transformer blocks (visual-audio streams) followed by $N_2$ unimodal transformer blocks (audio stream only). To ensure balanced semantic response across modalities, we propose an MMDiT structure that adopts dual-phase attentions, achieving audio-visual alignment through joint self-attention and subsequently implementing text semantic injection through cross-attention. Furthermore, to enhance audio generation quality, we leverage pre-trained audio representations to guide the modeling process through a representation alignment strategy.

The multimodal input representations in HiFi-Foley leverage specialized encoders to extract modality-optimized features: video frames are processed by a pre-trained SigLIP2 (Tschannen et al., 2025) encoder generating visual features $F_v \in \mathbb{R}^{L_v \times D_v}$. Textual descriptions are encoded through CLAP (Elizalde et al., 2023), yielding semantic embeddings $F_t \in \mathbb{R}^{L_t \times D_t}$. Audio waveforms are compressed via our enhanced DAC-VAE encoder into audio latents $x \in \mathbb{R}^{L_a \times D_a}$. Synchronization features $F_s \in \mathbb{R}^{L_s \times D_s}$ are extracted using Synchformer (Viertola et al., 2024), where $L_v$, $L_t$, $L_a$, and $L_s$ represent the correspondence of sequence lengths, and $D_v$, $D_t$, $D_a$, and $D_s$ represent the corresponding feature dimension.

### 3.1 MULTIMODAL ALIGNMENT WITH DUAL-PHASE ATTENTIONS.

TV2A generation requires modeling distinct alignment relationships between video and text modalities: audio and video exhibit fine-grained frame-level temporal and semantic dependencies, while audio-text interactions only rely on global semantic guidance. During our exploration of model architectures, we discover that triple-stream MMDiT structures tend to generate audio that heavily relies on textual content while neglecting visual information. In the triple-stream MMDiT, text and video are treated with equal importance, leading the model to preferentially learn from the high-density semantic information in text while overlooking the relatively sparse semantic content in video. We attribute this phenomenon to the fundamentally different alignment relationships that video and text modalities have with audio. This semantic response imbalance necessitates a more sophisticated attention design that accounts for the distinct characteristics of each modality. To address this dichotomy, our MMDiT architecture employs a dual-phase attention mechanism. Unlike conventional triple-stream MMDiTs that rely solely on joint self-attention for tri-modal fusion, our dual-phase approach differentiates between modalities by first establishing audio-visual alignment through joint self-attention, then separately incorporating textual semantics via cross-attention.

**Self-attention phase.** In the self-attention phase, audio latents and visual features are concatenated into a unified sequence, enhanced with interleaved rotary position embedding (RoPE). Traditional approaches apply RoPE (Su et al., 2024) to audio and visual sequences independently, which may not effectively capture the temporal correlations. Our interleaved RoPE strategy interleaves audio and visual tokens along the temporal dimension before applying position embeddings, thereby enabling the model to learn more coherent temporal relationships between visual-audio modalities. Specifically, given audio latents $x \in \mathbb{R}^{L_a \times D_a}$ and visual features $F_v \in \mathbb{R}^{L_v \times D_v}$. We first employ nearest-neighbor interpolation to align the sequence length to $L = max(L_a, L_v)$, then interleave two sequences. For timestep $t \in [1, L]$, the joint feature $F_{av}$ is combined through Equation 1:

$$\begin{cases} F_{av}[2t-1, :] = x[t, :] \\ F_{av}[2t, :] = F_v[t, :] \end{cases} \tag{1}$$

This operation creates an alternating pattern $F_{av}$ of audio and visual tokens. Subsequently, we apply RoPE to this interleaved sequence, ensuring that temporally adjacent audio and visual tokens receive consecutive position embeddings. Finally, we decouple the interleaved sequence back into separate audio latent $x'$ and visual feature $F'_v$, and then concatenate two sequences along the temporal dimension to serve as the query, key, and value in the following self-attention mechanism, shown in Equation 2.

$$F'_{av} = \text{SelfAttention}\left(\text{Concat}\left(x', F'_v\right)\right) \tag{2}$$

The aligned audio-visual features $F'_{av}$ are subsequently split into parallel processing streams, each transformed through linear projection layers equipped with gating mechanisms. These streams are then processed through adaLN layers that dynamically modulate audio latents and visual features. Both the gating mechanisms and modulation parameters utilize feature processing based on synchronization feature from the Synchformer visual encoder and flow timestep embeddings.

$$F_{out} = \text{CrossAttention}(Q = F'_{av}, K = F_t, V = F_t) \tag{3}$$

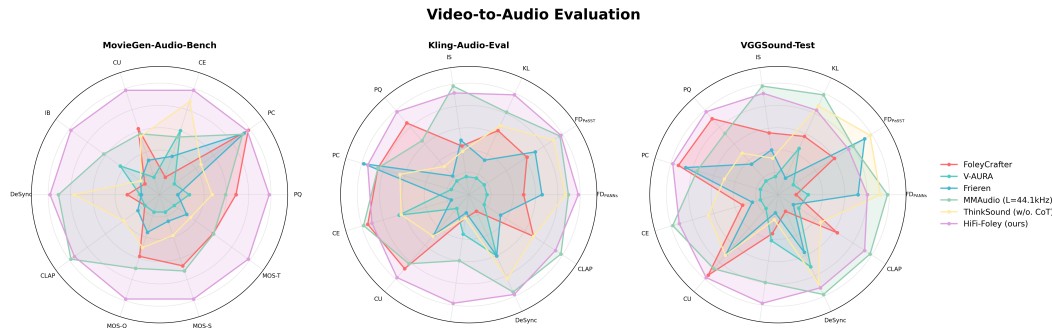

Figure 2: Radar Chart of Video-to-Audio Evaluation. It contains the results on three evaluation set: Kling-Audio-Eval, VGGSound-Test, and MovieGen-Audio-Bench, demonstrating that HiFi-Foley achieves comprehensive superiority.

**Cross-attention phase.** In the subsequent cross-attention phase, the concatenated audio-visual sequence $F'_{av}$ serves as the query, while CLAP-derived text embeddings $F_t$ provide the key and value components, shown in Equation 3, facilitating semantic alignment between multimodal inputs and textual descriptions. Following cross-attention, the enhanced features $F_{out}$ are decomposed into dual processing streams for audio and visual modalities. Each stream is subsequently processed through a transformation encompassing linear projection, gating mechanisms, adaLN, MLP operations, and final gating with residual connection.

## 3.2 TRAINING STRATEGY

The HiFi-Foley framework employs flow-matching as the primary training objective, which models the continuous transformation from noise to target audio representations. Given a source distribution $p_0$ (typically Gaussian noise) and target distribution $p_1$ (audio latents), flow-matching learns a vector field that defines the optimal transport path between these distributions. The flow-matching loss is formulated as:

$$\mathcal{L}_{\text{FM}}(\theta) = \mathbb{E}_{t,\mathbf{x}_0,\mathbf{x}_1} \left[ \|v_\theta(\mathbf{x}_t, t, \mathbf{c}) - u_t(\mathbf{x}_0, \mathbf{x}_1)\|^2 \right] \quad (4)$$

where $v_\theta$ is the learned vector field, $\mathbf{x}_t$ is the interpolated sample at time $t$, $\mathbf{c}$ denotes the conditioning information, and $u_t(\mathbf{x}_0, \mathbf{x}_1)$ represents the target vector field.

Additionally, we introduce the REPA strategy that involves aligning hidden states from intermediate layers of transformer blocks in our diffusion model with frame-level audio representations from the pre-trained ATST-Frame encoder. Through systematic ablation studies, we explored the impact of REPA loss placement across different transformer layers and found that applying REPA loss to the latent representations from the 8th unimodal transformer block yields optimal performance. Specifically, let $E_{ATST}$ denote a pre-trained ATST encoder, which produces representations $\mathbf{F_r} \in \mathbb{R}^{N \times D}$, where $\mathbf{F_r} = E_{ATST}(\mathbf{x})$, where $N$ representing the sequence length of audio feature and $D$ is the ATST feature dimension. REPA loss aims to align the mapped latents from the intermediate DiT layers $\mathbf{h_t} = f_\theta(\mathbf{z_t})$, where $\mathbf{z_t}$ denotes the audio latent compressed by a pretrained autoencoder and $f_\theta$ denotes trainable diffusion transformer layers, with the ATST-Frame audio features $\mathbf{F_r}$. This mapping is implemented through a Multi-Layer Perceptron (MLP) layer, denoted by $h_\phi$. Equation 5 shows the calculation of REPA loss.

$$\mathcal{L}_{\text{REPA}}(\theta, \phi) := -\mathbb{E}_{\mathbf{x}, \epsilon, t} \left[ \frac{1}{N} \sum_{n=1}^{N} \text{sim}\left( \mathbf{F_r}^{[n]}, h_\phi\left(\mathbf{h}_t^{[n]}\right) \right) \right] \quad (5)$$

The overall training objective combines both flow-matching and REPA losses with a weighting parameter $\lambda$, shown in Equation 6.

$$\mathcal{L}_{\text{total}}(\theta, \phi) = \mathcal{L}_{\text{FM}}(\theta) + \lambda \cdot \mathcal{L}_{\text{REPA}}(\theta, \phi) \quad (6)$$

This dual-objective training strategy leverages flow-matching loss to learn optimal generative trajectories while employing REPA loss to maximize semantic and acoustic alignment through cosine

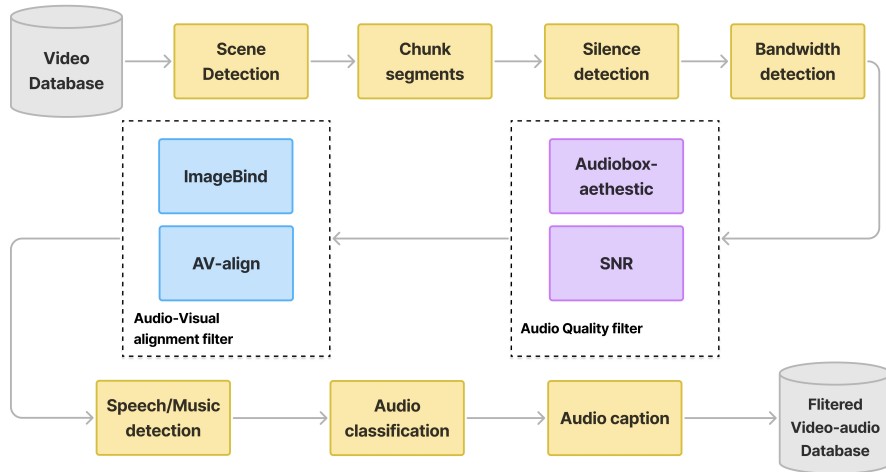

Figure 3: Data pipeline for filtering video-audio data. The workflow illustrates the processing steps from the raw video database to the filtered video-audio database.

Table 1: Comparison of Our Training Dataset with Existing Datasets.

| Datasets | Duration | Num. of Audios | Video | Audio | Audio Caption |
|---|---|---|---|---|---|
| AudioSet (Gemmeke et al., 2017) | 5.8kh | 2M | Yes | Yes | No |
| AudioCaps (Kim et al., 2019) | 416h | 150k | Yes | Yes | Yes |
| WavCaps (Mei et al., 2024a) | 7.6kh | 400k | No | Yes | Yes |
| VGGSound (Chen et al., 2020) | 556h | 200k | Yes | Yes | No |
| Epic Sounds (Huh et al., 2023) | 100h | 78.4k | Yes | Yes | No |
| Ours | 122kh | 55M | Yes | Yes | Yes |

similarity between pre-trained ATST features and DiT internal representations, collectively enhancing both generation fidelity and semantic coherence.

## 3.3 DATA CONSTRUCTION

The TV2A task presents a complex multimodal generation challenge that requires large-scale, high-quality text-video-audio datasets to produce robust and generalizable audio. Current open-source datasets, however, lack the necessary quality and scale to adequately support this demanding task. To bridge this gap, we develop a comprehensive data pipeline composed entirely of open-source tools and designed to systematically identify and exclude unsuitable content.

**Data Pipeline.** As illustrated in Figure 3, our multi-stage filtering process firstly eliminate videos lacking audio streams. Subsequently, we employ scene detection algorithms (Castellano, 2023) to segment raw videos, then chunk them into 8-second intervals. These segments undergo silence ratio analysis, with those exceeding an 80% silence threshold being discarded. Given the prevalence of heavily compressed and quality-degraded content on internet platforms, we implement bandwidth detection to ensure audio quality, retaining only samples with effective sampling rates exceeding 32 kHz. Audio quality constitutes a critical factor in generative audio tasks. Videos captured using substandard equipment often exhibit substantial background noise and ambient interference, rendering them unsuitable for generating cinematic-quality audio. To address this issue, we employ the Production Quality (PQ) metric from the AudioBox-aesthetic toolkit (Tjandra et al., 2025) for audio quality assessment. Additionally, signal-to-noise ratio (SNR) serve as a supplementary metric. To obtain a robust and accurate estimate, we employ the WADA-SNR (Kim & Stern, 2008) algorithm. Using these parameters, we empirically design a rigorous standard to filter and retain only high-quality audio segments. Specifically, we enforce a PQ score exceeding 6 on the AudioBox-Aesthetics scale and SNR threshold of $>-5$dB. Another challenge in the V2A domain is ensuring audio-video alignment, which consists of both semantic and temporal alignment. To address this, we leverage ImageBind (Girdhar et al., 2023) and AV-align (Yariv et al., 2024), retaining only samples with an ImageBind cosine similarity $>0.2$ and an AV-align score $>0.1$, respectively.

Table 2: Objective evaluation results on Kling-Audio-Eval. HiFi-Foley achieves superior performance across distribution matching ($FD_{PaNNs}$, KL), audio quality (PQ), visual-semantic alignment (IB) and temporal alignment (DeSync) metrics.

| Method | $FD_{PaNNs}\downarrow$ | $FD_{PaSST}\downarrow$ | KL↓ | IS↑ | PQ↑ | PC↓ | CE↑ | CU↑ | IB↑ | DeSync↓ | CLAP↑ |
|---|---|---|---|---|---|---|---|---|---|---|---|
| FoleyCrafter | 22.30 | 322.63 | 2.47 | 7.08 | 6.05 | 2.91 | 3.28 | 5.44 | 0.22 | 1.23 | 0.22 |
| V-AURA | 33.15 | 474.56 | 3.24 | 5.80 | 5.69 | 3.98 | 3.13 | 4.83 | 0.25 | 0.86 | 0.13 |
| Frieren | 16.86 | 293.57 | 2.95 | 7.32 | 5.72 | **2.55** | 2.88 | 5.10 | 0.21 | 0.86 | 0.16 |
| MMAudio (L-44.1kHz) | 9.01 | 205.85 | 2.17 | **9.59** | 5.94 | 2.91 | **3.30** | 5.39 | 0.30 | 0.56 | **0.27** |
| ThinkSound ($w/o.$ CoT) | 9.92 | 228.68 | 2.39 | 6.86 | 5.78 | 3.23 | 3.12 | 5.11 | 0.22 | 0.67 | 0.22 |
| HiFi-Foley (ours) | **6.07** | **202.12** | **1.89** | 8.30 | **6.12** | 2.76 | 3.22 | **5.53** | **0.38** | **0.54** | 0.24 |

Table 3: Objective evaluation results on VGGSound-Test. Our models achieves superior performance across audio quality (PQ) and visual-semantic alignment (IB).

| Method | $FD_{PaNNs}\downarrow$ | $FD_{PaSST}\downarrow$ | KL↓ | IS↑ | PQ↑ | PC↓ | CE↑ | CU↑ | IB↑ | DeSync↓ | CLAP↑ |
|---|---|---|---|---|---|---|---|---|---|---|---|
| FoleyCrafter | 20.65 | 171.43 | 2.26 | 14.58 | 6.33 | 2.87 | 3.60 | 5.74 | 0.26 | 1.22 | 0.19 |
| V-AURA | 18.91 | 291.72 | 2.40 | 8.58 | 5.70 | 4.19 | 3.49 | 4.87 | 0.27 | 0.72 | 0.12 |
| Frieren | 11.69 | 83.17 | 2.75 | 12.23 | 5.87 | 2.99 | 3.54 | 5.32 | 0.23 | 0.85 | 0.11 |
| MMAudio (L-44.1kHz) | **7.42** | 116.92 | **1.77** | **21.00** | 6.18 | 3.17 | **4.03** | 5.61 | 0.33 | **0.47** | **0.25** |
| ThinkSound ($w/o.$ CoT) | 8.46 | **67.18** | 1.90 | 11.11 | 5.98 | 3.61 | 3.81 | 5.33 | 0.24 | 0.57 | 0.16 |
| HiFi-Foley (ours) | 11.34 | 145.22 | 2.14 | 16.14 | **6.40** | **2.78** | 3.99 | **5.79** | **0.36** | 0.53 | 0.24 |

Table 4: Objective and subjective evaluation results on MovieGen-Audio-Bench. Our model achieves SOTA performance across almost all objective metrics and subjective evaluations.

| Method | PQ↑ | PC↓ | CE↑ | CU↑ | IB↑ | DeSync↓ | CLAP↑ | MOS-Q↑ | MOS-S↑ | MOS-T↑ |
|---|---|---|---|---|---|---|---|---|---|---|
| FoleyCrafter | 6.27 | **2.72** | 3.34 | 5.68 | 0.17 | 1.29 | 0.14 | 3.36±0.78 | 3.54±0.88 | 3.46±0.95 |
| V-AURA | 5.82 | 4.30 | 3.63 | 5.11 | 0.23 | 1.38 | 0.14 | 2.55±0.97 | 2.60±1.20 | 2.70±1.37 |
| Frieren | 5.71 | 2.81 | 3.47 | 5.31 | 0.18 | 1.39 | 0.16 | 2.92±0.95 | 2.76±1.20 | 2.94±1.26 |
| MMAudio (L-44.1kHz) | 6.17 | 2.84 | 3.59 | 5.62 | 0.27 | 0.80 | **0.35** | 3.58±0.84 | 3.63±1.00 | 3.47±1.03 |
| ThinkSound ($w/o.$ CoT) | 6.04 | 3.73 | 3.81 | 5.59 | 0.18 | 0.91 | 0.20 | 3.20±0.97 | 3.01±1.04 | 3.02±1.08 |
| HiFi-Foley (ours) | **6.59** | 2.74 | **3.88** | **6.13** | 0.35 | **0.74** | 0.33 | **4.14±0.68** | **4.12±0.77** | **4.15±0.75** |

Following the aforementioned filtering process, we annotate the remaining video segments using speech-music detection (Hung et al., 2022) and audio classification (Gong et al., 2021) models. These annotations provide categorical tags for each segment, enabling effective management of category distribution and ensuring balanced representation in the training dataset. Subsequently, we generate audio captions for each segment using GenAU (Haji-Ali et al., 2025), which provides concise descriptions of the audio content.

**Dataset.** Leveraging our proposed data pipeline, we have constructed a high-quality TV2A dataset. As shown in Table 1, our dataset is compared with five commonly used datasets. As illustrated, our dataset significantly surpasses all others in scale, comprising 122k hours of content with 55 million clips, each 8 seconds in length. Furthermore, our dataset provides comprehensive multi-modal resources, including video, audio, and audio captions. This large-scale, multi-modal collection offers robust support for enhancing the generalization capability and generation quality of our model.

## 4 EXPERIMENTS

### 4.1 EXPERIMENT SETTINGS

HiFi-Foley consists of 18 MMDiT layers and 36 unimodal audio DiT layers with a hidden dimension of 1536 and 12 attention heads. The training is conducted on 128 H20 GPUs with an effective batch size of 2048 over 200k steps on a 100k-hour TV2A datasets built by our proposed data pipeline, using the AdamW optimizer with a learning rate of 1e-4. We applied a classifier-free guidance (CFG) dropout rate of 0.1 for each modality. We employ DAC-VAE by removing the original residual vector quantization (RVQ) blocks from DAC. A detailed description is provided in the appendix.

We evaluate HiFi-Foley against existing SOTA models on three datasets: Kling-Audio-Eval, VGGSound-Test, and MovieGen-Audio-Bench. For objective evaluation, we employ a comprehensive multi-dimensional assessment covering distribution matching (FD and KL divergence using PANNs and PaSST), audio quality (IS and AudioBox-Aesthetics scores), visual-semantic alignment

(ImageBind cosine similarity), temporal alignment (Synchformer DeSync), and text-semantic consistency (LAION-CLAP). For subjective evaluation, we conduct MOS assessment on audio quality (MOS-Q), semantic alignment (MOS-S), and temporal alignment (MOS-T) using 527 MovieGen-Audio-Bench samples, with 20 experienced annotators providing ratings from 1 (poor) to 5 (excellent). For MOS-Q, scoring is performed using audio only, while for MOS-S and MOS-T, both audio and video are used for evaluation. For all reported results, the inference is performed using a classifier-free guidance (CFG) scale of 4.5 and 50 sampling steps.

## 4.2 MAIN RESULTS

**Kling-Audio-Eval.** Table 2 presents the objective evaluation results on the Kling-Audio-Eval dataset. HiFi-Foley demonstrates superior performance across multiple metrics, including distribution matching (FD, KL), audio quality (PQ), visual-semantic alignment (IB), and temporal synchronization (DeSync) in comparison with baselines. Compared with the current state-of-the-art model MMAudio, HiFi-Foley demonstrates slightly inferior performance on IS, CE, and CLAP scores, while achieving notable improvements in FD (9.01 to 6.07), KL (2.17 to 1.89), and IB (0.30 to 0.38) scores.

**VGGSound-Test.** The objective evaluation on the VGGSound-Test is shown in Table 3. Notably, HiFi-Foley underperforms some baselines in distribution matching metrics (FD, KL), but leads in audio quality metrics (PQ). This discrepancy may stem from the fact that most audio samples in VGGSound are recorded using non-professional equipment, resulting in generally poor audio quality that creates a substantial distribution gap with the outputs of HiFi-Foley. Nevertheless, our model maintains the SOTA performance in IB score while achieving comparable results in DeSync and CLAP metrics.

**MovieGen-Audio-Bench.** Table 4 displays both objective and subjective evaluation results on the MovieGen-Audio-Bench. HiFi-Foley exhibits outstanding generation quality, outperforming baselines in nearly all objective metrics and all subjective evaluations. Compared with the strong baseline MMAudio, our model demonstrates significant improvements across audio quality (PQ), temporal alignment (DeSync), and visual-semantic alignment (IB), while maintaining comparable performance in text-semantic alignment (CLAP).

As shown in the radar charts in Figure 2, comprehensive evaluation across all three datasets demonstrates that HiFi-Foley achieves substantial improvements in visual-semantic alignment (IB) over all baselines. Our model also leads in audio quality (PQ) and temporal alignment (DeSync) while maintaining competitive text semantic alignment (CLAP). In terms of distribution matching, HiFi-Foley achieves optimal performance on the Kling-Audio-Eval dataset. These results collectively demonstrate that HiFi-Foley establishes new state-of-the-art performance in TV2A generation.

## 4.3 ABLATION STUDY

Table 5: Ablation Study on Multimodal Transformer Block Architectures.

| Method | PQ ↑ | PC ↓ | CE ↑ | CU ↑ | IB ↑ | DeSync ↓ | CLAP ↑ |
|---|---|---|---|---|---|---|---|
| Joint self-attention | 6.32 | **2.72** | 3.61 | 5.76 | 0.31 | 1.05 | **0.32** |
| Parallel cross-attention | 6.33 | 2.80 | 3.59 | 5.55 | 0.26 | 0.81 | 0.27 |
| Joint self-attention+cross-attention | **6.38** | 2.76 | **3.68** | **5.90** | **0.32** | **0.78** | 0.30 |
| $w/o.$ interleaved RoPE | 6.36 | 2.78 | 3.65 | 5.77 | 0.31 | 0.79 | 0.30 |
| $w/o.$ unimodal DiT | 6.23 | 2.83 | 3.57 | 5.70 | 0.31 | 0.79 | 0.30 |

Table 6: Ablation Study on Representation Alignment Models.

| Method | PQ ↑ | PC ↓ | CE ↑ | CU ↑ | IB ↑ | DeSync ↓ | CLAP ↑ |
|---|---|---|---|---|---|---|---|
| $w/o.$ REPA | 6.23 | 2.83 | 3.57 | 5.63 | 0.31 | 0.79 | 0.30 |
| EAT+ATST | 6.00 | 2.90 | 3.54 | 5.43 | 0.32 | 0.79 | 0.29 |
| EAT only | 6.24 | 2.77 | 3.55 | **5.69** | 0.32 | 0.79 | 0.31 |
| ATST only | **6.28** | **2.74** | 3.59 | 5.68 | **0.33** | **0.75** | **0.33** |

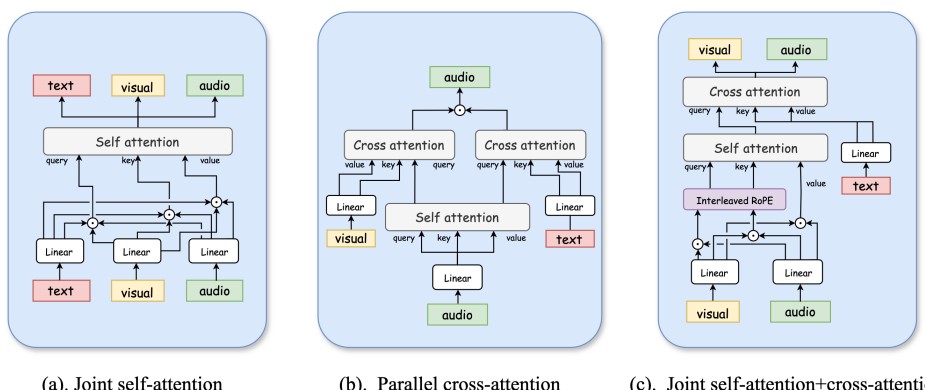

(a). Joint self-attention      (b). Parallel cross-attention      (c). Joint self-attention+cross-attention

Figure 4: Structural ablation study on multimodal attention mechanisms: (a) Triple-stream Joint Self-Attention. (b) Parallel Cross-Attention. (c) Dual-Phase Architecture by employing Interleaved RoPE and Joint Self-Attention for audio-video modeling before injecting text via Cross-Attention.

Table 7: Ablation Study on Representation Alignment for Multimodal and Unimodal Transformers.

| Method | PQ ↑ | PC ↓ | CE ↑ | CU ↑ | IB ↑ | DeSync ↓ | CLAP ↑ |
|---|---|---|---|---|---|---|---|
| MMDiT + UniDiT | 6.28 | 2.74 | 3.59 | 5.68 | 0.33 | 0.75 | **0.33** |
| MMDiT only | 6.28 | 2.79 | **3.70** | 5.73 | 0.33 | 0.79 | 0.32 |
| UniDiT only (Layer 8) | **6.34** | 2.80 | 3.67 | **5.77** | **0.34** | **0.74** | **0.33** |
| Layer 12 | 6.28 | **2.84** | 3.61 | **5.77** | 0.32 | 0.81 | 0.32 |
| Layer 16 | 6.32 | 2.74 | 3.58 | 5.75 | 0.33 | 0.78 | 0.32 |

To thoroughly investigate the impact of different model architectures on performance and validate the effectiveness of our proposed design, we conduct comprehensive ablation experiments on MovieGen-Audio-Bench. The ablation study primarily focuses on multimodal conditioning designs in MMDiT, the efficacy of the unimodal audio DiT, and optimal implementation strategies for representation alignment.

**Model Architecture.** For the architecture of MMDiT, we design two alternative experiments, shown in 4: (a) employing joint self-attention for text-audio-video triple-stream modal alignment, and (b) using parallel cross-attention to separately align audio-text and audio-video modals. All configurations maintain identical experimental setups with excluding REPA and employing unimodal DiT. As shown in Table 5, the proposed approach, which first achieves audio-video alignment through joint attention, and then injects text features through cross-attention to the audio-video sequence, outperforms alternatives across most metrics, particularly demonstrating significant improvement in temporal alignment (DeSync). Additionally, when replacing interleaved-RoPE strategy with conventional RoPE, we observe performance degradation across metrics, confirming that interleaved RoPE effectively enhances audio-video modality alignment. To verify the effectiveness of the unimodal transformer, we further replace unimodal DiT with audio-video dual-stream DiT. The results show that the audio-only transformer achieved superior performance compared with the replacement approach.

**Representation Alignment.** For representation alignment, we compare two widely-used pre-trained audio self-supervised models: EAT (Chen et al., 2024) and ATST (Li et al., 2023). Table 6 reveals that using ATST yields the best results, with noticeable improvements in audio quality, temporal alignment, and text-semantic alignment. Notably, combining EAT and ATST leads to performance degradation across most metrics, attributable to the divergence in feature distributions between the two models, which prevents them from providing robust guidance. Furthermore, we investigate the effects of applying REPA in different stages and layers. The results in Table 7 show that REPA achieves optimal performance when applied in unimodal DiT, with additional observations suggesting better outcomes when applied to shallower layers of the unimodal blocks.

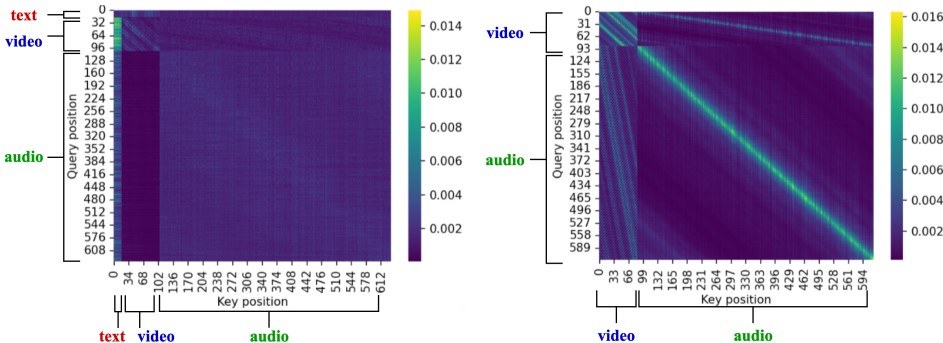

Figure 5: Visualization of Attention Maps. Left (Triple-Stream Baseline): Audio latent overwhelmingly attend to text latent (vertical stripes), ignoring visual cues. Right (Ours): A distinct diagonal pattern emerges in the audio-visual attention block, indicating that audio latent are attending to the video latent.

### 4.4 DISCUSSION

**Balanced Visual and Textual Semantics.** The structural innovation of HiFi-Foley stems from its strategic use of distinct attention mechanisms for visual and textual feature injection. This approach effectively addresses the issue of generated audio relying excessively on text semantics while overlooking video semantics. The experiments show that HiFi-Foley achieves superior performance across visual-semantic alignment (IB) with maintaining competitive text-semantic alignment (see Section 4.2 and 4.3), which reveals that joint self-attention is particularly effective for aligning video features with strong temporal correspondence to audio, whereas separate cross-attention better processes text features that convey global contextual information.

To validate this mechanism, we visualize representative attention maps of the inference process, shown in Figure 5. In triple-stream architectures, we observe a distinct phenomenon of "attention collapse", where audio latent overwhelmingly attend to text embeddings while exhibiting sparse and disordered activation on visual latent. This confirms that without structural constraints, the model tends to exploit semantic shortcuts from text. In contrast, our dual-phase architecture induces a clear diagonal alignment pattern between audio queries and visual keys. This physical evidence demonstrates that our decoupled design forces the model to establish fine-grained correspondence with visual dynamics before integrating global textual semantics, thereby structurally resolving the modality imbalance.

**Enhanced Audio quality Through REPA Strategy and Dataset Scaling.** HiFi-Foley significantly improves the quality of video-to-audio generation by introducing the REPA training strategy. This approach effectively aligns the hidden representations of DiT with robust self-supervised features. Additionally, our proposed data pipeline facilitates the scalable construction of our large-scale and high-quality training dataset, further enhancing the model performance.

## 5 CONCLUSION

In this work, we introduce HiFi-Foley, a novel text-video-to-audio generation framework that integrates dual-phase attention mechanisms within MMDiTs alongside a representation alignment training strategy. This approach enables high-fidelity audio synthesis with well-balanced alignment to both visual semantics and textual context, as well as precise audio-visual temporal synchronization. Additionally, we develop an efficient data pipeline based on open-source tools, offering scalable support for the construction of high-quality TV2A datasets. Extensive experiments demonstrate that HiFi-Foley sets a new state-of-the-art performance in text-video-to-audio generation, with notable strengths in video-semantic alignment, temporal synchronization, and overall audio quality.

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

# A EXPERIMENT DETAILS

## A.1 VARIATIONAL AUTOENCODER.

The proposed DAC-VAE adopts a neural audio codec architecture based on variational autoencoders, specifically designed for high-fidelity audio compression and reconstruction tasks at 48kHz. The model achieves efficient compressed representation and accurate reconstruction of audio signals through an encoder-decoder framework.

### A.1.1 ENCODER.

The encoder employs a multi-scale hierarchical downsampling structure to extract hierarchical feature representations by progressively reducing temporal resolution. Starting with a 64-dimensional feature representation, a total temporal compression ratio of 960 is achieved using a downsampling factor sequence of [2, 3, 4, 5, 8]. Each encoder layer consists of three residual units and one downsampling convolutional layer. The residual units incorporate skip connections and integrate two Snake activation functions and two weight-normalized convolutional layers. The first convolutional layer uses a 7×1 kernel with multi-scale dilated convolutions (dilation rates of 1, 3, and 9) to capture audio features at different temporal scales, while the second convolutional layer employs a 1×1 kernel for feature fusion. Compared to the traditional ReLU activation function, the Snake activation function demonstrates superior performance in modeling periodic signals. Downsampling is implemented via strided convolution, with the kernel size set to twice the downsampling factor, while the number of feature channels is doubled to maintain representational capacity.

### A.1.2 LATENT SPACE MODELING.

The model utilizes a continuous latent space representation instead of traditional discrete codebook quantization. A quantized convolutional layer maps the 128-dimensional features output by the encoder to a 256-dimensional output, where the first 128 dimensions represent the mean parameters of a Gaussian distribution and the latter 128 dimensions represent the log-variance parameters, thereby constructing a diagonal Gaussian distribution. This continuous representation design enables smoother feature interpolation and improved generation quality by avoiding the information loss associated with discrete quantization.

### A.1.3 DECODER.

The decoder adopts a symmetric upsampling structure relative to the encoder, starting with an initial feature dimension of 1536 and progressively restoring the original 48kHz audio temporal resolution through an upsampling factor sequence of [8, 5, 4, 3, 2]. Before decoding begins, a post-quantization convolutional layer remaps the 128-dimensional variables sampled from the latent distribution into a feature space suitable for decoding. Each decoding module consists of one transposed convolutional upsampling layer and three residual units, maintaining the same multi-scale dilated convolution design as the encoder to ensure feature consistency. Upsampling is implemented via transposed convolution, with the kernel size set to twice the upsampling factor, while the number of feature channels is halved. The decoder employs the snakebeta activation function and enables logarithmic-scale parameterization to enhance numerical stability during computation.

### A.1.4 EXPERIMENT SETTINGS.

Our DAC-VAE is trained on approximately 100k hours of audio data for 700k steps using 32 NVIDIA H20 GPUs with a batch size of 256. We adopt the AdamW optimizer with a learning rate of 1e-4 for optimization. The implemented system operates at a sampling rate of 48kHz, with a latent vector dimensionality of 128 and a latent rate of 50Hz.

### A.1.5 AUDIO RECONSTRUCTION.

For audio reconstruction, we conduct comparative studies between DAC (Kumar et al., 2023) and the continuous VAE employed in Stable Audio Open (Evans et al., 2024). The evaluation spanned three distinct domains: AudioSet for general sounds, Song Describer for music, and LibriTTS-Clean

testset for speech scenarios. We adopt Perceptual Evaluation of Speech Quality (PESQ), Short-Time Objective Intelligibility (STOI), Scale-Invariant Signal-to-Distortion Ratio (SI-SDR) and Mel distance as objective metrics. As shown in Table 8, our proposed DAC-VAE achieves superior performance across all metrics on three evaluation sets. These experiments validate that our DAC-VAE delivers robust reconstruction performance across diverse audio domains, establishing its effectiveness as a general-purpose audio reconstruction framework.

Table 8: Evaluation of autoencoder reconstructions in AudioSet (Sound), Song Describer (Music) and LibriTTS (Speech).

| Dataset | Method | Sample rate | PESQ↑ | STOI↑ | SI-SDR↑ | Mel-dist↓ | Latent rate | Latent |
|---|---|---|---|---|---|---|---|---|
| AudioSet | DAC | 44.1kHz | 4.17 | 0.94 | 11.08 | 0.48 | 86Hz | discrete |
| | Stable Audio Open | 44.1kHz | 2.33 | 0.72 | 3.32 | 0.83 | 21.5Hz | 64-dim |
| | DAC-VAE (ours) | 48kHz | 3.59 | 0.91 | 8.41 | 0.60 | 50Hz | 64-dim |
| | DAC-VAE (ours) | 48kHz | **4.45** | **0.98** | **14.76** | **0.27** | 50Hz | 128-dim |
| Song Describer | DAC | 44.1kHz | 4.18 | 0.96 | 13.84 | 0.48 | 86Hz | discrete |
| | Stable Audio Open | 44.1kHz | 2.56 | 0.83 | 8.02 | 0.79 | 21.5Hz | 64-dim |
| | DAC-VAE (ours) | 48kHz | 3.57 | 0.93 | 12.60 | 0.57 | 50Hz | 64-dim |
| | DAC-VAE (ours) | 48kHz | **4.45** | **0.99** | **17.40** | **0.29** | 50Hz | 128-dim |
| LibriTTS Clean Set | DAC | 44.1kHz | 4.29 | 0.98 | 12.37 | 0.47 | 86Hz | discrete |
| | Stable Audio Open | 44.1kHz | 2.68 | 0.93 | 5.78 | 0.87 | 21.5Hz | 64-dim |
| | DAC-VAE (ours) | 48kHz | 3.75 | 0.97 | 9.51 | 0.61 | 50Hz | 64-dim |
| | DAC-VAE (ours) | 48kHz | **4.50** | **0.99** | **14.37** | **0.26** | 50Hz | 128-dim |

## A.2 BANDWIDTH TAGGING.

To address the varying sampling rates in our training data, we introduce a bandwidth tagging strategy. Audio samples with sampling rates above 16 kHz receive a "high-quality" tag in their captions. During inference, we correspondingly append this tag to all input captions. Our experiments demonstrate that this method successfully conditions the model to associate the "high-quality" tag with higher sampling rates, resulting in audio outputs with enhanced high-frequency detail preservation. This bandwidth-aware conditioning significantly improves spectral fidelity, as shown by the superior high-frequency retention in the generated waveforms.

## A.3 EVALUATION METRICS.

We adopt a comprehensive suite of metrics spanning multiple dimensions:

- **Distribution Matching**
  - **Fréchet Distance (FD)**: Measures the similarity between generated and real audio feature distributions using mean and covariance statistics (lower values indicate better alignment), computed using PANNs and PaSST embeddings.
  - **Kullback-Leibler Divergence (KL)**: Quantifies probability distribution divergence between generated and real audio features through PANNs.
- **Audio Quality**
  - **Inception Score (IS)**: Evaluates quality and diversity through the PANNs classifier.
  - **AudioBox-Aesthetics**:
    * *Production Quality (PQ)*: Focuses on the technical aspects of quality instead of subjective quality. Aspects including clarity & fidelity, dynamics, frequencies and spatialization of the audio;
    * *Production Complexity (PC)*: Focuses on the complexity of an audio scene, measured by number of audio components. In our experiments, we found that audio with significant noise and unnatural artifacts tend to receive higher PC scores, whereas clean, human-perceptually pleasant audio samples are assigned lower scores. Therefore, in the context of Foley generation, we argue that lower PC scores are preferable, as they indicate reduced noise and closer alignment with human auditory perception;

- ∗ *Content Enjoyment (CE)*: Focuses on the subject quality of an audio piece. It's a more open-ended axis, some aspects might includes emotional impact, artistic skill, artistic expression, as well as subjective experience, etc;
- ∗ *Content Usefulness (CU)*: Also a subjective axis, evaluating the likelihood of leveraging the audio as source material for content creation.

- **Visual-Semantic Alignment**
  - **ImageBind (IB) Cosine Similarity**: Measures cross-modal alignment between video frames and generated audio embeddings using ImageBind's joint embedding space (higher scores indicate better alignment).

- **Temporal Alignment**
  - **DeSync**: Predicts audio-visual synchronization errors via Synchformer (lower values indicate tighter temporal coherence).

- **Text-Semantic Consistency**
  - **LAION-CLAP Score**: Measures semantic similarity between input text and generated audio through LAION-CLAP (higher scores reflect better textual grounding).

- **Subjective Evaluation**
  - **MOS-Q**: The acoustic quality and auditory naturalness of the generated audio, independent of video content (1-5 scale);
  - **MOS-S**: The degree of matching between the category, source characteristics, and physical attributes of the generated audio with the content depicted by the video frames and textual semantics (1-5 scale);
  - **MOS-T**: The accuracy of synchronization between the generated audio and visual events, including onset/offset timing and duration (1-5 scale).

- **Audio Reconstruction**
  - **PESQ**: Perceptual Evaluation of Speech Quality (1-4.5 scale)
  - **STOI**: Short-Time Objective Intelligibility (0-1)
  - **SI-SDR**: Scale-Invariant Signal-to-Distortion Ratio (dB)
  - **Mel Distance**: Distance between ground-truth and generated Mel-spectrograms

## A.4 BASELINE DETAILS

In our experimental setup, we conduct comprehensive comparisons with five baseline models: FoleyCrafter ( two-stage generation), V-AURA (autoregressive method), Frieren (first flow-matching method), MMAudio (current SOTA model), and ThinkSound (latest related work). To ensure fair comparisons, all models are evaluated using their officially released pre-trained versions, with inference performed on identical hardware configurations and following the original inference scripts. When multiple pre-trained variants are available, we consistently select the best version for benchmarking. Notably, for ThinkSound, we only evaluate the version without Chain-of-Thought (CoT) instructions due to the unavailability for the pre-trained LLM component responsible for generating CoT instructions. A brief introduction to these baselines follows:

- **FoleyCrafter**: A TV2A framework that ensures audio generation through a pretrained text-to-audio model, featuring a semantic adapter with cross-attention for visual relevance, and a temporal controller with onset detection for precise synchronization.

- **V-AURA**: The first autoregressive video-to-audio model achieving fine-grained alignment via high frame-rate visual features and cross-modal fusion.

- **Frieren**: A V2A model based on rectified flow matching for spectrogram generation via ODE sampling. Employs transformer-based cross-modal fusion for alignment.

- **MMAudio**: A TV2A framework jointly trained on video-audio and text-audio data to enhance semantic alignment. Uses flow matching and a frame-level sync module for efficiency, achieving SOTA performance in previous works.

- **ThinkSound**: Integrates Chain-of-Thought reasoning into a three-stage pipeline: foundational Foley generation, interactive object-centric refinement, and language-guided editing.

# B VISUALIZATION

## B.1 TRAINING DATASET.

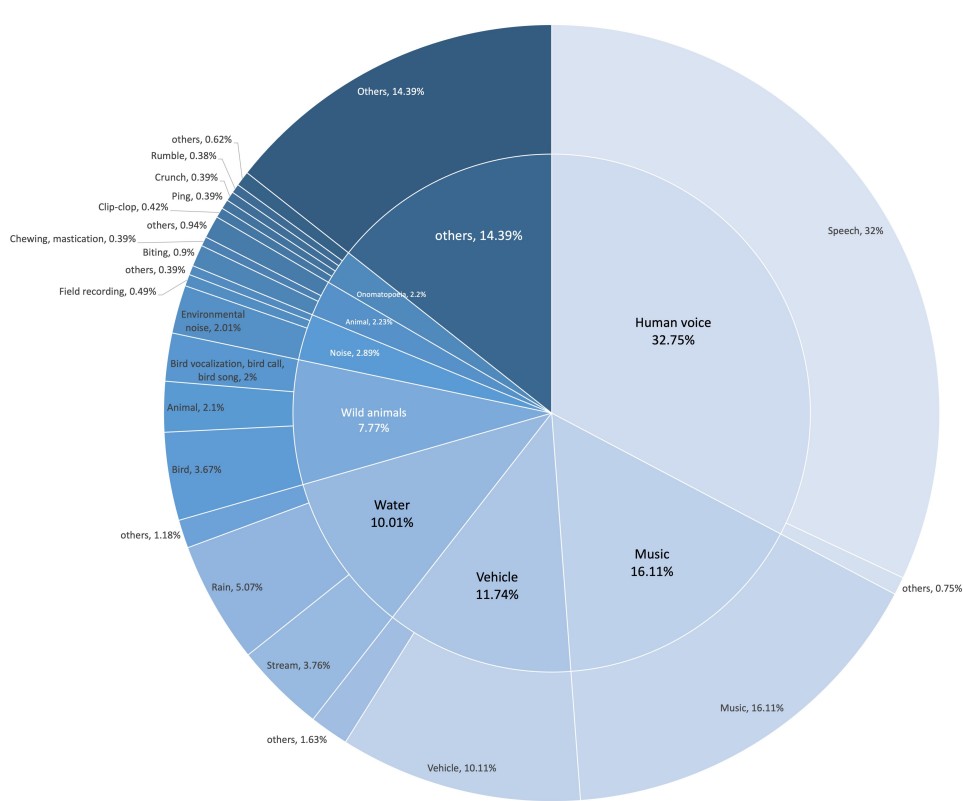

Figure 6: Composition of Sound Event Categories within the Training Dataset.

We visualize the distribution of major sound categories in the training dataset, as illustrated in Figure 6. Through our proposed efficient data pipeline, we have constructed a large-scale, high-quality video–audio-text dataset comprising 122k hours of sounding videos along with corresponding audio captions. The dataset encompasses a wide variety of real-world acoustic scenes, including human activities, music, vehicles, natural environments, and animal vocalizations, establishing a solid foundation for the generalization and robust generative capabilities of HiFi-Foley.

## B.2 ATTENTION MAPS.

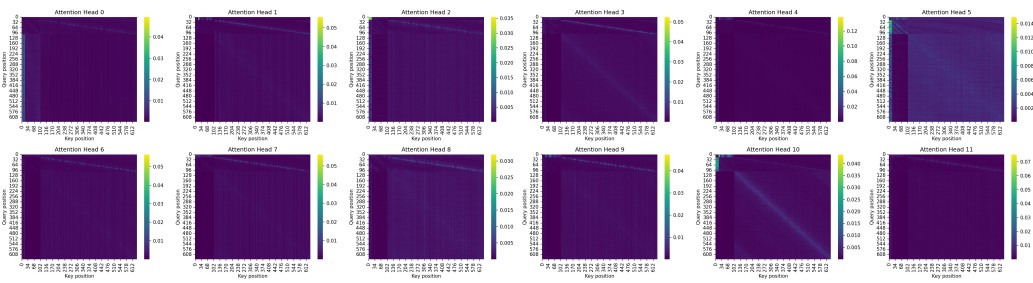

Figure 7: Visualization of the Triple-Stream Attention Map at Inference Step 0, MM-DiT Block 0.

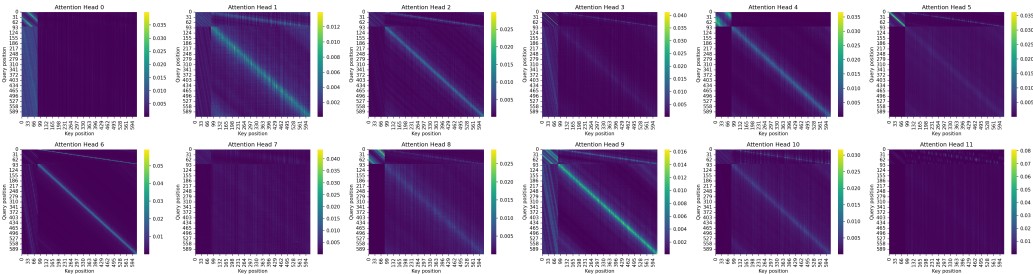

Figure 8: Visualization of the Dual-Stream Attention Map at Inference Step 0, MM-DiT Block 0.

### B.3 Spectrogram.

We present spectrogram visualization between our method and existing approaches in Figure 9 and 10. Notably, our method demonstrates stable performance in preserving high-frequency components without spectral leakage, while maintaining precise temporal alignment between audio events and corresponding actions.

## C Limitations and Future Work

### C.1 Limitations

Although our method achieves state-of-the-art performance, several limitations remain. First, while our model architecture effectively mitigates the issue of text semantic dependency, the model's response to video semantics is still insufficient, failing to comprehensively capture all elements present in the video. Second, the audio captions from GenAU are relatively brief and contain certain hallucinations, leading to omission and confusion of audio elements. Additionally, when processing longer prompts, the generated audio tends to include background music, although this music often aligns well with the visual content. This phenomenon may occur because audio captions containing music are typically longer, while pure sound effect captions are shorter, causing the model to assume background music should be present when encountering lengthy prompts. This issue can be mitigated through negative prompting techniques. Third, our model cannot yet generate intelligible speech and can only produce indiscernible vocalizations when faced with scenes involving human dialogue.

### C.2 Future Work

In future work, we will further explore more effective multimodal diffusion architectures to better address the problem of multimodal condition competition. We plan to design an audio captioning model that integrates both audio and video content, which will significantly enhance our video-to-audio generation model's semantic responsiveness. Additionally, we will extend the model's capabilities to include speech and fine-grained music generation, ultimately achieving unified audio generation capabilities.

## D The Use of Large Language Models

In this work, large language models (LLMs) are employed solely for the purpose of linguistic refinement and grammatical correction of the manuscript. The model does not contribute to the conceptualization, analysis, or intellectual content of the research. All ideas, arguments, and conclusions remain entirely those of the authors.

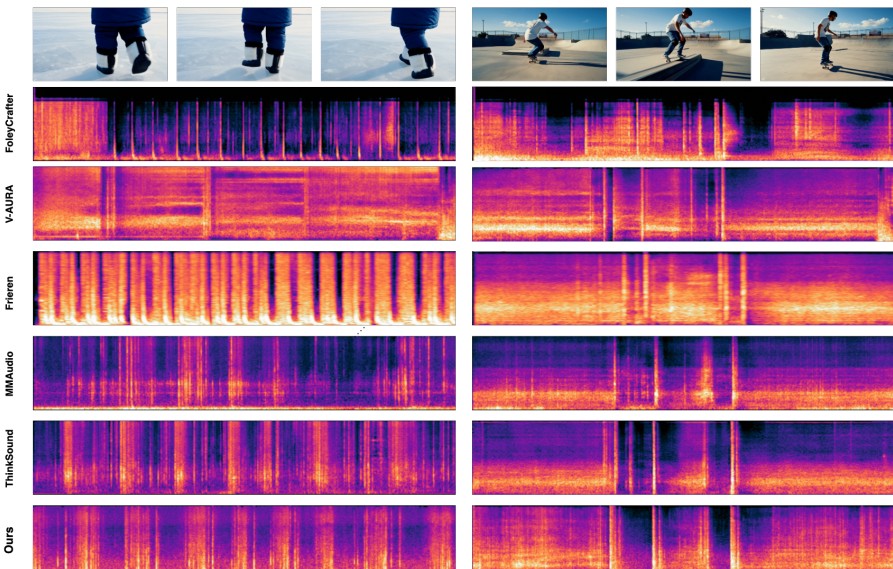

Figure 9: Left: The video sequence illustrates a walking scenario on icy surfaces, where our proposed method achieves precise temporal alignment for both the initiation/termination timing and the duration of each step. Right: Spectral analysis confirms accurate synchronization with the temporal characteristics of human movements in the skateboarding scenario.

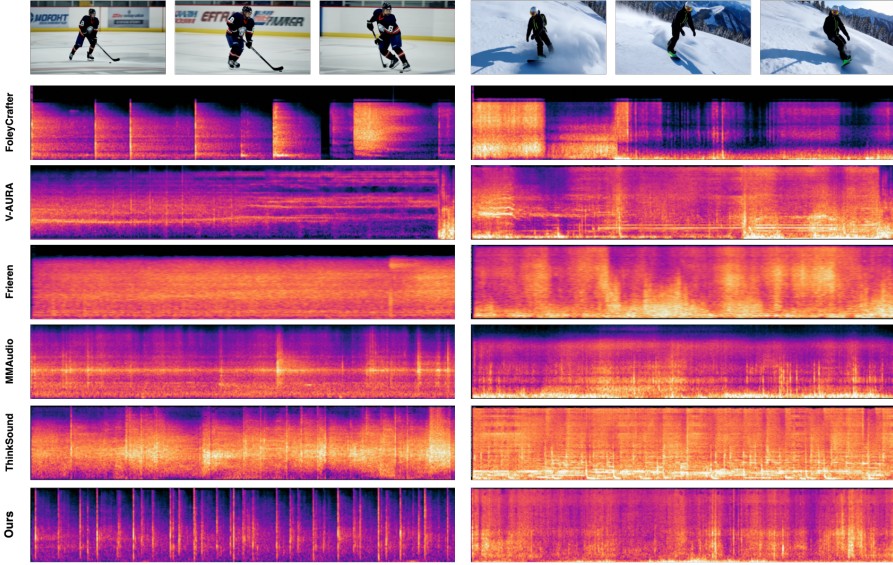

Figure 10: Left: In the ice hockey scenario involving rapid rhythmic auditory cues, our spectral analysis demonstrates robust performance in detecting subtle motion variations synchronized with the sound patterns. Right: Our method preserves the full spectral representation in complex skiing scenario where motion-sound alignment is less distinct, with no discernible degradation of high-frequency components in the spectrogram.

