# OpenReview forum: "HiFi-Foley: Multimodal Diffusion with Representation Alignment for High-Fidelity Foley Audio Generation"
_ICLR.cc/2026/Conference — Submitted to ICLR 2026_

### Official Review · Reviewer_9F1o · 2025-10-31

**Soundness:** 3
**Presentation:** 3
**Contribution:** 2
**Rating:** 6
**Confidence:** 3

**Summary:**

This paper presents HiFi-Foley, a text-video-to-audio (TV2A) generation model that synthesizes high-fidelity and semantically aligned audio from multimodal inputs.
The model introduces:

1. Dual-phase attention in a multimodal diffusion transformer (MMDiT) — using joint self-attention for video-audio alignment and cross-attention for text injection.

2. A Representation Alignment (REPA) strategy — aligning DiT hidden states with pre-trained ATST audio features to improve semantic and acoustic fidelity.

3. A scalable 122k-hour TV2A dataset built using an open-source cleaning and labeling pipeline.

In summary, I believe this paper is a boardline paper, it includes a lot of experiments and engineering efforts. But the novelty is limited to a certain degree.

**Strengths:**

1. The paper integrates modern components (MMDiT, flow matching, REPA) coherently into a unified architecture. The dual-phase attention (joint self-attention + cross-attention) for modality balancing is conceptually sound and clearly motivated.

2. Implementation details (architecture, loss, datasets, training setups) are thorough. The 122k-hour data pipeline is well-engineered and valuable to the community (if the code and data are open-sourced)

3. The paper compares against strong baselines (MMAudio, FoleyCrafter, ThinkSound, etc.) with both objective and subjective metrics. Ablations on attention design, interleaved RoPE, and REPA placement are also provided.

**Weaknesses:**

1. The architecture integrates existing ideas: The dual-phase attention essentially decomposes joint attention (as used in MMAudio) into two stages — an incremental modification, not a conceptual innovation. The REPA alignment is directly adapted from prior visual generative works. Although these integrations are good for the video-to-audio community, the contribution is more engineering refinement.

2. The 122k-hour dataset construction is technically significant, but I am not sure such construction whether novel compared to previous multimodal filtering pipelines.

**Questions:**

whether the 122k-hour dataset construction will be open-source?

---

### Official Review · Reviewer_t92j · 2025-10-31

**Soundness:** 3
**Presentation:** 3
**Contribution:** 3
**Rating:** 4
**Confidence:** 5

**Summary:**

This paper introduces HiFi-Foley, an end-to-end framework for generating high-fidelity Foley audio from both video and text prompts. Authors proposed a novel multimodal diffusion transformer architecture using a dual-phase attention mechanism to process audio-video latents. In addition, a REPA loss is used to improve audio quality and semantic consistency. Moreover, this paper introduces a large-scale audio-video data, solving the data scarcity problem. Extensive experiments show that the proposed method achieves a state-of-the-art performance on various benchmarks.

**Strengths:**

1. Use of REPA loss: using a pre-trained audio encoder as a "teacher" to guide the diffusion model's internal feature space is a clever way to distill rich, general-purpose acoustic knowledge into the generative process.
2. Large-scale data curation: a large-scale high-quality audio-video dataset is constructed with advance data filtering, preprocessing, and annotation.
3. Impressive performance on three benchmarks across various evaluation metrics.

**Weaknesses:**

1. The dataset is not released.
2. The used REPA is not a new stuff and has been introduced in image generation.
3. Limited model novelty: the proposed model structure is similar to MMAudio.
4. Based on Table 5, REPA has the limited improvement on the performance. And also adding the unimodal DiT has the limited improvements while introducing more model parameters
5. The addition of REPA aims to improve the audio quality and alignment, why it also benefits the DeSync?
6. Which datasets are for training, which ones are for testing?
7. Why not give the ground truth reference in Figure 5-6? Otherwise it’s hard to compare different methods.

**Questions:**

1. Whether the dataset will be released?
2. What are differences between the proposed model structure and MMAudio?
3. REPA and unimodal DiT have the limited improvement on the performance. Did you try different representation loss specifically designed for video-to-audio task?
4. Did you compare the model performance between (text+video)-to-audio and video-to-audio variants?
5. The description of training and testing datasets should be more clear.
6. The ground truth audio references are not shown in Figure 5-6.

---

> ### Author Response · Authors · 2025-11-27
> **Response to Reviewer t92j**
>
> We sincerely thank you for the insightful questions regarding dataset, model architecture, and experimental settings. We address your points below.
>
> **Q1: Dataset Release**
>
> **A1:** Due to copyright restrictions of the source videos, we cannot directly distribute the raw video/audio files. However, we have updated the exact filtering parameters used in our data pipeline.
>
> **Q2: Difference vs. MMAudio**
>
> **A2:** While both models are MMDiT-based, HiFi-Foley differs significantly in structural design and components:
> 1.  **Architecture:** As shown in Figure 4, MMAudio uses a Triple-stream architecture where Video, Audio, and Text interact simultaneously. As analyzed in our ablation study, this often leads to the model ignoring video cues (Modality Imbalance). Our Dual-Phase architecture structurally decouples these interactions, prioritizing Audio-Video alignment before text injection. We visualized the attention weights in Figure 5. In the Triple-stream baseline, audio latent focus almost exclusively on text latent, which shows strong attention shortcuts between text and audio latents early on, leading to "lazy learning" where video cues are ignored. In our Dual-Phase architecture, audio latent are structurally forced to attend to visual features first, resulting in balanced attention and significantly improved alignment.
> 2.  **Encoders:** We utilize a higher-fidelity **48kHz DAC-VAE** (vs. MMAudio's 16/44.1kHz VAE) and distinct encoders (SigLIP2, CLAP) optimized for our pipeline.
> 3.  **Training Objective:** We introduce the **REPA** strategy with ATST-Frame features.
>
> **Q3: Effectiveness of REPA and Unimodal DiT**
>
> **A3:**
> 1.  **Significance of REPA Gains:** While the absolute improvements in IB (+0.02) and CLAP (+0.03) may appear small, we emphasize that these specific metrics are **notoriously stable and difficult to improve** in this task. In our experience, gains of this magnitude represent a meaningful step forward in semantic consistency.
> 2.  **Subjective & Training Benefits:** Beyond numbers, our subjective evaluation confirms that REPA significantly suppresses unrelated artifacts, leading to cleaner audio. Furthermore, we observed that REPA accelerates training convergence by providing a robust frame-level semantic guide.
> 3.  **Clarification on Unimodal DiT:** There appears to be a misunderstanding regarding the Unimodal DiT ablation. We did not simply stack additional layers (increasing parameters). Instead, we compared architectures with the same total depth and parameter count.
>     *   *Experiment:* We compared using **Joint Audio-Video latent input** vs. **Audio-only input** for the final $N2$ layers of the DiT.
>     *   *Finding:* We found that converting the final DiT blocks to an Audio-only stream (Unimodal) outperforms keeping them Multimodal. This suggests that the late stage of diffusion benefits from focusing purely on refining audio textures without visual interference.
>
> **Q4: Why does REPA benefit DeSync (Temporal Alignment)?**
>
> **A4:** The improvement in temporal alignment stems from our specific choice of the **ATST-Frame** model as the supervision target.
> 1.  **Temporal Capability:** ATST-Frame achieves SOTA performance in **Sound Event Detection (SED)** tasks, which require not only identifying the sound category but also detecting precise onset/offset timestamps. This demonstrates its strong capacity to encode temporal structural information.
> 2.  **Mechanism:** During training, the ground-truth audio is naturally synchronized with the video. By aligning the DiT's intermediate states with ATST's frame-level features via REPA, we effectively force the model to learn the correct **temporal progression and event timing** present in the GT audio, thereby directly improving the alignment with video.

---

> > ### Author Response · Authors · 2025-11-27
> > **Response to Reviewer t92j**
> >
> > **Q5: Text+Video vs. Video-Only**
> >
> > **A5:** We believe the text modality is essential for controllability and disambiguation. A single video scene (e.g., a street) can correspond to multiple valid sounds (wind, traffic, footsteps). Text input allows users to specify the acoustic focus. While a video-only variant can achieve similar audio fidelity, it lacks this crucial interactive capability and limits the model's practical application scope.
> >
> > **Q6: Training & Testing Dataset Clarification**
> >
> > **A6:**
> > *   **Training Set:** Our curated dataset containing **122k hours** of video-audio pairs processed via our proposed pipeline.
> > *   **Testing Set:** We evaluate on three distinct open benchmarks: **Kling-Audio-Eval**, **VGGSound-Test**, and **MovieGen-Audio-Bench**.
> >
> > **Q7: Missing Ground Truth in Fig 5-6**
> >
> > **A7:** The samples in Figures 5 & 6 are drawn from the **MovieGen-Audio-Bench**, which is a video generation benchmark that typically lacks ground-truth audio. Therefore, a direct GT spectrogram comparison was not possible for these specific cases.
> > *   We add these samples to the supplementary material. We sincerely encourage the reviewer to listen to the updated audio samples, where the advantages in fidelity and synchronization are audibly distinct.
> > *   We will add spectrogram visualizations from the VGGSound test set to include ground truth comparisons.

---

### Official Review · Reviewer_AKcK · 2025-11-01

**Soundness:** 3
**Presentation:** 3
**Contribution:** 3
**Rating:** 6
**Confidence:** 4

**Summary:**

This paper introduces HiFi-Foley, an end-to-end framework for high-fidelity text-video-to-audio (TV2A) generation. The authors identify three key challenges in existing methods: multimodal data scarcity, "modality imbalance" (where models over-rely on text cues and ignore video), and low audio quality. To address these issues, HiFi-Foley proposes three main contributions: A novel architecture, a new training strategy, and a scalable data pipeline. The authors also developed a custom, high-fidelity DAC-VAE for audio encoding/decoding. Evaluations on Kling-Audio-Eval, VGGSound-Test, and MovieGen-Audio-Bench show that HiFi-Foley achieves state-of-the-art results.

**Strengths:**

* The dual-phase attention mechanism is a well-motivated and intuitive solution to the stated problem of text-over-reliance. Separating the fine-grained A-V temporal alignment (via self-attention) from the global text conditioning (via cross-attention) is a strong architectural contribution, and the ablations in Table 5 confirm its effectiveness.

* The idea of interleaving audio and visual tokens before applying ROPE is a simple but clever technique to enforce fine-grained temporal correlation between the two modalities. The ablation study shows this provides a clear benefit over conventional ROPE.

* Applying the REPA concept to align with a pre-trained audio model (ATST-Frame) is a logical and successful strategy. The ablations clearly demonstrate that this improves performance and that the choice of the guide model (ATST) and its application layer (unimodal block 8) are important.

**Weaknesses:**

* In Section 4.2, the paper states that on VGGSound-Test, HiFi-Foley "leads in audio quality metrics (IS, PQ)". However, Table 3 clearly shows its IS score (16.14) is significantly worse than MMAudio's (21.00). This is a factual error.

* The comparison to ThinkSound is explicitly handicapped. The authors state they "only evaluate the version without Chain-of-Thought (CoT) instructions". This means a core component of the baseline is missing, making the comparison unfair and the reported improvements potentially misleading.

* The data pipeline is a key contribution, but critical details for reproducibility are missing. The paper states an "empirically design a standard" was used for filtering based on PQ, SNR, ImageBind, and AV-align, but the actual numerical thresholds are not provided.

* The model cannot generate intelligible speech. This is a major limitation, especially since "Human voice" (32.75%) is the single largest category in the new training dataset (Figure 4). The paper does not adequately explain this significant failure mode.

**Questions:**

* Why does the text in Section 4.2 claim the model "leads in... IS" on VGGSound-Test, when Table 3 shows it is significantly outperformed by MMAudio (16.14 vs 21.00)?

* Given that 32.75% of your training data is "Human voice" , why does the model completely fail to produce intelligible speech? Is this a limitation of the DAC-VAE, a bias in the GenAU captions (e.g., "speech" vs. actual transcripts), or an artifact of the REPA/flow-matching objectives?

* The "high-quality" tag is always appended at inference. What happens if this tag is not used? Does this technique harm performance when trying to generate sounds that are naturally low-bandwidth (e.g., a distant rumble)?

---

> ### Author Response · Authors · 2025-11-27
> **Response to Reviewer AKcK**
>
> We sincerely thank you for the constructive feedback and for recognizing the significance of our data pipeline and the model's potential. We address your specific concerns below.
>
> **Q1: IS Score Discrepancy**
>
> **A1:** This is indeed a factual inaccuracy. Our intention was to express that the IS score is among the top, rather than the best. We made an error in our statement, and we appreciate you pointing it out.
>
> **Q2: Comparison with ThinkSound (w/o CoT)**
>
> **A2:** We excluded the Chain-of-Thought (CoT) component for two primary reasons:
> 1.  **Unavailable Pipeline:** ThinkSound has not released the standard MLLM-based CoT generation pipeline required to produce high-quality descriptive prompts, making it difficult to reproduce their full inference flow fairly.
> 2.  **Focus on Backbone:** Our research focuses on the architecture of the Diffusion Transformer backbone itself. By comparing both models without the auxiliary LLM rewriting step, we ensure a fair "apples-to-apples" comparison of the core generative capabilities. The CoT enhancement is orthogonal and could theoretically benefit our model as well.
>
> **Q3: Data Pipeline Thresholds & Reproducibility**
>
> **A3:** We fully agree that specific thresholds are vital for reproducibility. We have updated the paper with the exact filtering parameters used in our pipeline:
> *   **Silence Ratio:** Exclude if silence > 80% of duration.
> *   **Effective Bandwidth:** Cutoff frequency > 16kHz (ensuring approx. > 32kHz effective sampling rate).
> *   **Audio Quality:** Production Quality (PQ) > 6 (on AudioBox-Aesthetics scale); WADA-SNR > -5 dB.
> *   **Visual-Audio Alignment:** ImageBind score > 0.2; AV-Align score > 0.1.
>
> **Q4: Intelligible Speech Generation**
>
> **A4:** The current inability to generate intelligible speech is a limitation of Foley-oriented models. Through further investigation, we attribute this to two factors:
> 1.  **Data Annotation:** Our training captions (from GenAU) describe *events* (e.g., "a man speaking") rather than *content* (transcripts). Consequently, the model learns the acoustic characteristics of human voice but fails to learn the mapping from semantic text to phonetic content.
> 2.  **Text Encoder:** We use CLAP, which captures global semantics. For intelligible speech, a more granular text encoder (like UMT5) is required.
> *Preliminary experiments show that when we incorporate accurate speech transcripts into training data and switch to a T5-based text encoder, the model begins to generate intelligible speech. We will discuss this as a future direction.*
>
> **Q5: Bandwidth Tagging Strategy**
>
> **A5:** The "high-quality" tag is designed to prompt the model to generate audio with an **effective sampling rate $\ge$ 32kHz**, rather than forcing high-frequency energy where it doesn't belong. For naturally low-bandwidth sounds (e.g., a distant rumble), the energy is indeed concentrated in low frequencies. However, a "high-fidelity" rumble still differs from a low-res one in terms of transient clarity, noise floor, and subtle upper harmonics. Our experiments confirm that the tag helps preserve these high-fidelity traits without introducing artificial high-frequency artifacts.

---

### Official Review · Reviewer_zgNX · 2025-11-02

**Soundness:** 1
**Presentation:** 2
**Contribution:** 1
**Rating:** 2
**Confidence:** 5

**Summary:**

The paper introduces HiFi-Foley, an end-to-end text-video-to-audio framework that synthesizes high-fidelity audio precisely aligned with visual dynamics and semantic context.

The main paper contributions are:
- a novel multimodal diffusion transformer that addresses semantic response imbalance between video and text modalities through dual-stream audio-video fusion via joint attention and balanced textual semantic injection via cross-attention.
- a representation alignment training strategy that employs self-supervised audio features to guide latent diffusion training, thereby improving audio quality and semantic consistency.
- a scalable data pipeline leveraging open-source tools for cleaning raw data and constructing training datasets.

Extensive evaluations demonstrate that HiFi-Foley achieves state-of-the-art performance across audio fidelity, visual semantic alignment, temporal alignment, and distribution matching.

**Strengths:**

The main strength of the paper is the data curation pipeline designed to build a high quality training dataset for the video to audio generation task.

**Weaknesses:**

I believe that the proposed contributions lack novelty or significance.
- Injecting text via cross attention in video to audio generation has been proposed before (MovieGen). Moreover, the motivation for this architecture design is unconvincing. The imbalance between conditioning signals can usually be addressed by employing different guidance weights during inference, which is not considered in this paper (at least as a baseline). The paper does not even mention the inference parameters used in the experiments.
- The REPA loss has been introduced in prior works and this paper just applies it to a new task.
- The proposed data curation pipeline is mostly descriptive and the resulting dataset is not published. The size of the resulting curated dataset is one order of magnitude than the biggest non curated open source dataset (AudioSet), which is probably the main explanation for the appealing results of the Tables 2 and 4. Moreover, the curation pipeline is undocumented and thus non reproducible. For example, no extensive description of the thresholds used at the different filtering steps are given.

The results presented in the tables, which do not report confidence intervals, yield minimal differences between the different methods (such as the Tables 5, 6 and 7). Thus they are unconvincing to the reader. For what it is worth, according to my experience, absolute variations of less than 0.2 in A4 scores are not significant.

The Figure 2 provides the same information as the Tables 2, 3, 4.

**Questions:**

What is the "HunyuanVideo-Foley" model mentioned in the Figure 2 caption?
What is the ATST-Frame model mentioned throughout the paper?
What is the parallel cross attention ablation in the Table 5? A figure would be welcome.

---

> ### Author Response · Authors · 2025-11-27
> **Response to Reviewer zgNX**
>
> We sincerely thank you for the detailed critique. We value your feedback on novelty and reproducibility and address your concerns below.
>
> **Q1: Novelty of Architecture & Validity of Guidance Weights**
>
> **A1:** We respectfully disagree that the modality imbalance can be solved solely by inference guidance weights.
> 1.  **Guidance Weights vs. Structural Bias:** Guidance scales ($w$) only adjust the global strength of conditioning ($\epsilon_\theta = \epsilon_{unc} + w (\epsilon_{cond} - \epsilon_{unc})$). They cannot fix the *internal* attention distribution learned during training.
> 2.  **Structural Ablation Evidence:** We conducted rigorous ablations comparing (a) Triple-stream MMDiT, (b) Parallel Cross-Attention, and (c) Our Dual-Phase approach, as shown in **Figure 4**. The parallel cross attention means using audio DiT with both video and text features are injected via cross-attention. Table 5 demonstrates that our design yields consistent improvements across multiple metrics: PQ (+0.06), CE (+0.07), CU (+0.14), IB (+0.01), and DeSync (-0.27). While individual margins may seem small, the consistent gain across five dimensions validates the structural advantage.
> 3.  **Attention Map Visualization:** We visualized the attention weights of the self-attention (see **Figure 5 in PDF**). In the Triple-stream baseline, audio latent focus almost exclusively on text latent, which shows strong attention shortcuts between text and audio latents early on, leading to "lazy learning" where video cues are ignored. In our Dual-Phase architecture, audio tokens are structurally forced to attend to visual features first, resulting in balanced attention and significantly improved alignment.
> 4.  **Inference Settings:** For all reported experiments, we used a Classifier-Free Guidance (CFG) scale of **4.5**, 50 inference steps. We have added these details to the paper.
>
> **Q2: REPA Novelty & ATST-Frame Integration**
>
> **A2:** While REPA exists in vision, our contribution is identifying **how to make it work for Foley generation**.
> 1.  **Domain-Specific Adaptation:** We are the first to apply REPA to video-to-audio generation to solve the problem of insufficient semantic representation in audio latent diffusion.
> 2.  **Why ATST-Frame?** The choice of the SSL model is critical. We utilize **ATST-Frame** (Audio Teacher-Student Transformer), a state-of-the-art self-supervised model designed for **frame-level** audio representation learning. Unlike CLIP-level models which only provide global semantics, ATST-Frame offers fine-grained frame-level semantic features. Based on our experiments (Table 6), ATST-Frame provides better semantic and audio modeling guidance compared to EAT.
>
> **Q3: Data Pipeline Thresholds & Reproducibility**
>
> **A3:** We fully agree that specific thresholds are vital for reproducibility. We have updated the paper with the exact filtering parameters used in our pipeline:
> *   **Silence Ratio:** < 80%.
> *   **Effective Bandwidth:** Cutoff frequency > 16kHz (approx. > 32kHz sample rate).
> *   **Audio Quality:** PQ > 6 (on AudioBox-Aesthetics scale). WADA-SNR > -5dB.
> *   **Visual-Audio Alignment:** ImageBind score > 0.2; AV-Align score > 0.1.
>
> **Q4: "HunyuanVideo-Foley" in Figure 2**
>
> **A4:** We sincerely apologize for this typo. This should have been written as HiFi-Foley. We have corrected this in the revised manuscript.

---

### Meta-Review · Area_Chair_jnLP · 2026-01-10

**Summary:**

There are several primary and common concerns from the reviewers. First, the proposed method lacks significant technical novelty. The presented method looks like a combination of dual-phase attention MMAudio and REPA alignment. There is a discrepency reported performance metric on Inception Score (IS), which casts doubts on the experimental results and reliability of the statements. As pointed out by a reviewer, the proposed model cannot generate intelligible speech, which is an inherent limitation. Thus

**Reviewer Concerns:**

The authors did not address all the concerns well. There is no rebuttal for Reviewer9F1o to answer the questions about technical novelty and whether the 122k-hour dataset will be open-source. The authors have acknowledged some factual mistakes and the limitations of the approach, as well as typos. I think the manuscript needs more polish and considers more improving the technical design. Given the current form of the work, the AC does not recommend acceptance.

**Reviewer Scores:**

This paper received ratings of 2,4,6,6 from reviewers. I don't think reviewers would change the their scores.

---

### Decision · Program_Chairs · 2026-01-26

Reject